# Preliminary Investigation of Side Effects of Polymyxin B Administration in Hospitalized Horses

**DOI:** 10.3390/antibiotics12050854

**Published:** 2023-05-05

**Authors:** Julia N. van Spijk, Katrin Beckmann, Meret Wehrli Eser, Martina Stirn, Andrea E. Steuer, Lanja Saleh, Angelika Schoster

**Affiliations:** 1Equine Department, Vetsuisse Faculty, University of Zurich, Winterthurerstrasse 260, CH-8057 Zurich, Switzerland; 2Department of Small Animals, Vetsuisse Faculty, University of Zurich, Winterthurerstrasse 260, CH-8057 Zurich, Switzerland; 3Departement for Clinical Diagnostics and Services, Vetsuisse Faculty, University of Zurich, Winterthurerstrasse 260, CH-8057 Zurich, Switzerland; 4Institute of Forensic Medicine, University of Zurich, Winterthurerstrasse 190/52, CH-8057 Zurich, Switzerland; 5Institute of Clinical Chemistry, University Hospital Zurich, Rämistrasse 100, CH-8091 Zurich, Switzerland

**Keywords:** neurotoxicity, ataxia, endotoxemia, nephrotoxicity

## Abstract

Neuro- and nephrotoxicity of polymyxins are known but clinical studies in horses are lacking. The aim of this study was to describe neurogenic and nephrogenic side effects of hospitalized horses receiving Polymyxin B (PolyB) as part of their treatment plan. Twenty horses diagnosed with surgical colic (*n* = 11), peritonitis (*n* = 5), typhlocolitis (*n* = 2), pneumonia, and pyometra (each *n* = 1) were included. Antimicrobial treatment was randomized to GENTA (gentamicin 10 mg/kg bwt q24 h IV, penicillin 30.000 IU/kg q6 h IV) or NO GENTA (marbofloxacin 2 mg/kg bwt q24 h IV, penicillin 30.000 IU/kg q6 h IV). The duration of PolyB treatment ranged from 1 to 4 days. Clinical and neurological examinations were performed, and serum PolyB concentrations were measured daily during and three days following PolyB treatment. Urinary analysis, plasma creatinine, urea and SDMA were assessed every other day. Video recordings of neurological examinations were graded by three blinded observers. All horses showed ataxia during PolyB treatment in both groups (median maximum ataxia score of 3/5, range 1–3/5). Weakness was detected in 15/20 (75%) horses. In 8/14 horses, the urinary γ-glutamyltransferase (GGT)/creatinine ratio was elevated. Plasma creatinine was mildly elevated in 1/16 horses, and SDMA in 2/10 horses. Mixed-model analysis showed a significant effect of time since last PolyB dose (*p* = 0.0001, proportional odds: 0.94) on the ataxia score. Ataxia and weakness should be considered as reversible adverse effects in hospitalized horses receiving PolyB. Signs of tubular damage occurred in a considerable number of horses; therefore, the nephrotoxic effect of polymyxins should be considered and urinary function monitored.

## 1. Introduction

Endotoxemia is common in horses with colic, diarrhea, sepsis, and other bacterial infections [1,2,3,4]. Polymyxin B (PolyB) is a peptide antibiotic used in these patients due to its endotoxin binding effects [5,6]. Several studies showed a decrease in proinflammatory cascade activation and clinical improvement in endotoxemic horses treated with PolyB [7,8,9,10,11]. PolyB is therefore an important therapeutic agent to control endotoxemia-associated disease and complications include laminitis, coagulopathy, renal and multi-organ failure [12].

Toxicity of polymyxins is well described in humans and restricts their use despite excellent antimicrobial activity especially against Gram-negative bacteria including some multi-drug-resistant species [13,14,15,16]. Nephrotoxicity is the most common and most important adverse effect of polymyxin treatment in human patients [13,16,17]. Accumulation in proximal tubule cells results in renal dysfunction and failure [18]. Factors increasing the risk for nephrotoxicity include the duration and dose of therapy, concomitant administration of other nephrotoxic medications, and pre-existing diseases [13]. Neurotoxicity, although less common, leads to paresthesia, dizziness, ataxia, and possibly life-threatening apnea in human patients [13,19]. Recent studies in mice indicate mitochondrial dysfunction in neurons, but pathophysiology is incompletely understood and other mechanisms such as a reduced sensitivity of the motor end plate to acetylcholine or calcium-dependent prolongation of neuronal depolarization have been proposed [14,20]. Factors influencing neurotoxicity include the duration and dose of therapy, concomitant medications, and pre-existing disease such as myasthenia gravis [13].

Studies on adverse effects of PolyB in horses are sparse. Measurement of the urinary γ-glutamyltransferase (GGT)/creatinine ratio was performed as a marker of renal damage in several studies evaluating the anti-endotoxemic effect of this drug and no increase was seen [9,10,21]. The presence of neurological side effects has not been reported in these publications; however, some preliminary data on a weak, ataxic gait after PolyB administration in endotoxemic horses exists [22]. Recently, we demonstrated the development of reversible ataxia and weakness in healthy horses receiving PolyB. The number of doses and co-administration of gentamicin were identified as risk factors [23]. At our hospital, PolyB is routinely used in horses at risk for or suffering from endotoxemia and similar clinical observations have been made.

The objective of this study was to describe the incidence, characteristics, severity, and duration of neurogenic and nephrogenic side effects after PolyB administration in hospitalized horses. Possible influencing factors such as the duration of treatment, PolyB serum concentrations, and co-administration of gentamicin were evaluated. The hypothesis was that all horses treated with PolyB show neurological adverse effects such as ataxia or weakness and that severity depends on duration of therapy, and co-administration of gentamicin, which would worsen the adverse effects.

## 2. Results

### 2.1. Animals

Twenty-four horses fulfilling the inclusion criteria were initially included in this study. Horses lacking video recordings were excluded; therefore, a total of 20 horses were finally included in this study. There was an equal distribution of mares and geldings (10/20, 50% each). Breeds included a mixed population of Warmbloods (*n* = 6), Arabians (*n* = 3), Icelandics (*n* = 3), Spanish Horses (*n* = 2), Shetlandponies (*n* = 2), Thoroughbreds (*n* = 1), and Draft horses (including Haflinger, Freiberger, Tinker, each *n* = 1). The age ranged from 2 to 26 years, with a median of 10.5 years.

Included horses were diagnosed with surgical colic (*n* = 11; large colon volvolus 4/11, left dorsal displacement of the colon 1/11, colon ascendens impaction 2/11, strangulating colon descendens lesion 2/11, and strangulating small intestinal lesion 2/11), peritonitis (*n* = 5; aseptic of unknown reason 3/4, septic due to vaginal tear 1/4, septic due to rectal tear 1/4), typhlocolitis of unknown cause (*n* = 2), aspiration pneumonia (*n* = 1), and pyometra (*n* = 1).

The duration of PolyB treatment ranged from 1 to 4 days (median 2 days) and 2 to 8 doses (median 4 doses). There were 8 horses in group 1 (GENTA, receiving gentamicin and penicillin) and 12 horses in group 2 (NO GENTA, receiving marbofloxacin and penicillin). This antimicrobial treatment was administered for 1–7 days (median 4 days). In 9 horses, antimicrobial treatment was continued orally thereafter with trimethoprim sulfonamides (5/25 mg/kg bwt q12hr PO, ROTA TS ad us. vet., Vetoquinol AG Bern; 3/9 horses), marbofloxacin (3 mg/kg bwt q12hr PO, Forcyl 160 mg/mL ad us. vet., Vetoquinol AG Bern; 6/9 horses), or metronidazol (25 mg/kg bwt q12hr PO, metronidazol 1000 mg/g compounded by the Tierspitalapotheke Zurich; 2/9 horses). In all but one horse, this change occurred after PolyB treatment was stopped. The total duration of antimicrobial treatment was 1–21 days (median 4 days).

Other medications administered included flunixin (18/20, 90%), intravenous fluids (18/20, 90%), dalteparin (15/20, 75%), lidocaine (CRI up to 24 h duration, 5/20, 25%), omeprazole (3/20, 15%), heparin (2/20, 10%), metamizole (2/20, 10%), sucralfate (2/20, 10%), fenbendazole (1/20, 5%), pergolide (1/20, 5%), and meloxicam (1/20, 5%). Sedative drugs were administered intravenously in 11/20 (55%) horses during the study period including xylazine (9/20, 45%), detomidin (4/20, 20%), and butorphanol (6/20, 30%); neurological exams were performed at least 2 h after administration of sedative drugs.

Hematology and clinical chemistry results of individual horses can be found in Appendix A.

SIRS was present in 14/20 (70%) horses, with 6/20 (30%) at a score of 2/4, 4/20 (20%) at a score of 3/4, and 4/20 (20%) at the maximum score of 4/4.

### 2.2. Neurological Effects

No horse had to be withdrawn from this study due to a live ataxia score ≥ 4/5.

A total of 75 neurological exams were video recorded and analyzed. Horses were video recorded at least once (1 video per horse in 2/20 horses, 3 videos in 4/20, 4 videos in 9/20, and 5 videos in 5/20).

All horses showed ataxia grade 1–3/5 in both groups (see Figure 1). Median maximum ataxia score was 3/5 (range 1–3/5) with 10/20 horses having their highest ataxia score during PolyB treatment (day 1 to day 3), 8/20 on the first day (post 1), 1/20 horses on the third day after PolyB treatment (post3), and in one horse ataxia was stable (grade 1/5) before and during PolyB treatment. Median ataxia scores were 2/5 (1–3/5) on day 1, 2.5/5 (1–3/5) on day 2, 2/5 (range 1–3/5) on day 3, 2/5 (range 1–3/5) on day post 1, 2/5 (range 0–3/5) on day post 2, and 1/5 (range 0–3/5) on day post 3. Ataxia scores of individual animals over time can be found in Appendix A.

Weakness was detected at least once in 15/20 horses (75%) and in 26/75 video exams (35%). Weakness was most common on day 3 (2/3 horses, 67%), followed by day 2 (4/8 horses, 50%), and day post 1 (8/17 horses, see Figure 2).

Statistical analysis showed a significant effect of time since last PolyB dose (*p* = 0.0001, effect size: 0.48) on the daily ataxia score. The proportional odds of this effect on the ataxia score was 0.94 (CI 0.91–0.97). In other words, for every hour after the last dose of PolyB, the ataxia score decreases by 6% per hour or by 99% (CI 89–99.9%) per 72 h.

No other factors were statistical significantly associated with the daily ataxia score of horses. None of the evaluated factors was significantly associated with the maximum ataxia score of horses.

The intra-rater agreement of the ataxia grading was good, with weighted kappa values of 0.55, 0.74, and 1.0. Inter-rater agreement was low, with a Krippendorf alpha value of 0.59 (CI 0.51–0.67).

### 2.3. Kidney Function Testing Results

Creatinine and urea were measured in 16/20 horses on day 1 and in 2/3 horses on day 3. Creatinine was elevated in 1/16 (6%) horses (176 μmol/L) on day 1, but within the refence range in the same horse on day 3. In all other horses, creatinine and urea measurements were within the reference range.

Urine analysis was performed in 14/20 horses on day 1. Urine specific gravity was <1.025 in 12/14; all 12 horses received intravenous fluid therapy at time of sampling. In 7/14 (50%) horses, altered urinary creatinine/serum creatinine ratio was measured; all 7 horses received intravenous fluid therapy and had a urine specific gravity of ≤1.010 at time of sampling. None of the 14 sampled horses showed proteinuria. Sediment analysis showed the presence of erythrocytes (4–8/HPF) in 2/14 (14%) horses and the presence of leukocytes in 2/14 (14%) other horses. Of these 4 horses, 2 had underwent laparotomy with placement of a urinary catheter the day before urine analysis. A low number of hyaline casts was detected in 1/14 (7%) horses in absence of proteinuria (UPC of 0.12 g/L). The urinary GGT/creatinine ratio was elevated in 8/14 (57%) horses on day 1 (see Figure 3).

SDMA was measured in 19/20 horses on day 1. Mildly elevated values (18 μg/dL and 19 μg/dL, reference value < 15 μg/dL) were found in 2/19 (11%) horses.

### 2.4. Polymyxin B Concentrations

Median serum PolyB concentrations were 0.0 μg/mL (range 0.0–0.5 μg/mL) on day 0 before start of PolyB treatment (Prae/Day 0) and 2.7 μg/mL (range 0.9–6.2 μg/mL) after the first dose (Post/Day 0), 4.7μg/mL (range 2.7–7.8 μg/mL) on day 1, 4.5 μg/mL (range 0.9–7.9 μg/mL) on day 2, 4.9 μg/mL (range 3.2–5.0 μg/mL) on day 3, and 0.0 μg/mL (range 0.0–0.0 μg/mL) on all 3 days after discontinuing PolyB treatment. Serum PolyB concentrations per group are shown in Figure 4.

### 2.5. Gentamicin Concentrations

Serum gentamicin concentrations were only measured in horses in group 1 (GENTA) during PolyB treatment. Median serum gentamicin concentrations were 6.1 mg/L (range 1.1–12.8 mg/L) on day 1, 13.8 mg/L (0.4–24.2 mg/L) on day 2, and 0.4 mg/L (0.4–19.9 mg/L) on day 3.

## 3. Discussion

This study showed the presence of neurogenic side effects in all hospitalized horses treated with PolyB. Signs included weakness and mild to moderate ataxia which decreased after the last dose. Nephrogenic side effects included signs of mild renal tubular damage in a considerable number of horses receiving PolyB as part of their treatment plan.

Neurotoxicosis due to polymyxin administration is known in human medicine causing signs such as paresthesia, dizziness, ataxia, and apnea [13,19]. In healthy horses, we recently showed the development of ataxia and weakness due to PolyB administration [23]. Similarly, preliminary results in equine patients and reports in calves showed the development of such clinical signs [22,24]. All horses in the present study showed neurological side effects such as ataxia and weakness, which is in accordance with these previous studies. The overall rate of neurotoxicity of 3% in human patients is, however, much lower than in this study, where all horses showed neurological abnormalities. Species specific susceptibility to toxicities or different detection rates of neurological signs are possible explanations. Subtle ataxia might be easier recognized in standing horses and due to their larger size compared to sick humans. Gait abnormalities might go underrecognized in hospitalized horses when animals are box rested during treatment but might also be overlooked in severely sick human patients.

Paresthesia is the most common sign of neurotoxicosis in humans and abnormal sensatory pathways might also explain ataxia in horses [13,19]. The pathomechanism of PolyB neurotoxicity is incompletely understood; while earlier study suggested a neuromuscular blockade by a reduced sensitivity of the motor end plate to acetylcholine, newer studies show neuronal oxidative stress and mitochondrial dysfunction in response to polymyxin administration in mice [14,20,25]. Signs of neuromuscular blockade include muscle weakness up to life threatening apnea, while involvement of sensory pathways would typically lead to paresthesia and/or ataxia. All such clinical signs are seen in polymyxin-induced neurotoxicosis and both pathways may play a role simultaneously. Further research is needed to understand underlying pathomechanisms and possible differences between species.

In this study, we detected a statistically significant negative effect of the time since the last dose on ataxia gradings displaying a decrease in severity after therapy was stopped. Severity of neurological deficits depends on the dose and duration of polymyxin administration and clinical signs are reversible after discontinuation of therapy in humans and healthy horses [13,19,23]. Further influencing factors on the development of neurological signs include other medications, primary disease, and pre-existing neurological abnormalities [13]. Co-administration of other neurotoxic drugs, for example aminoglycosides or calcium channel blocker is a known risk factor for amplification of neuromuscular blockade [26,27]. In healthy horses, we recently demonstrated an influence of gentamicin co-administration on ataxia severity during PolyB administration [23]. In the present study, there was no such effect shown, although we included two groups of antimicrobial treatment to evaluate this. Due to licensing problems, the used PolyB product was withdrawn from the market and the desired number of horses needed to show the calculated difference in ataxia grades between groups was not reached. An influence of gentamicin co-administration in a similar population in a more powerful study is therefore possible. Neurotoxicity of lidocaine is known in horses and leads to central nervous agitation shown as nervousness, agitation, sedation, ataxia, or collapse. These signs are reported to rapidly resolve when the treatment is stopped [28,29]. In the present study, only 5/20 horses were treated with systemic lidocaine for the first 24 h concurrently to PolyB, but all 20 horses showed neurological signs during and/or after treatment. Statistical analysis did reveal no effect of lidocaine co-administration on the maximum ataxia gradings, but the power of this analysis was weak due to low numbers. It seems very unlikely that seen ataxia was due to lidocaine instead of PolyB administration, but a possible influence of this drug on the nervous system pathology cannot be excluded. Ataxia is also described as a side effect of alpha-2 agonists, such as xylazine and detomidine [30]. These drugs are only short acting and neurological exams were performed before or at least two hours after their administration. Clinical signs such as visible ataxia are usually not expected after this time interval; however, the effect of residual serum concentrations is unknown and an influence on ataxia gradings cannot be completely excluded. Furthermore, the possible influence of other administered drugs is unclear. In vitro studies showed synergistic interaction of polymyxin with a range of non-antibiotic drugs [31]. The use of omeprazole prevented polymyxin E-induced nephrotoxicity in a rat model and was used in some horses concurrently to PolyB [32]. Metamizol on the other hand is reported to cause acute kidney injury in a low number of human patients and might have also influenced renal function in horses concurrently treated with this antipyretic drug in the present study [33]. No data exist on the influence of other concurrently used drugs such as anti-inflammatory, anticoagulant, and gastroprotective substances on the pharmacokinetics, therapeutic and side effects of PolyB. Interactions between these drugs and PolyB therefore cannot be excluded.

Presence of pre-existing neurological abnormalities was not reported in any of the included horses. However, in most horses in this study no neurological exam was performed before PolyB administration because most horses were presented after-hour in an emergency setting. Other possible influencing factors associated with the primary disease of sick horses such as changes in serum protein, albumin, and calcium concentrations or grade of systemic inflammatory response (SIRS score) were evaluated, but no influencing factors were identified. The power of this analysis was low and these and other factors, such as subclinical neurologic or renal disease, breed or individual predisposition, cannot be excluded. No association between PolyB serum concentration and the severity of neurological signs was found, which is in accordance with previous studies in healthy horses and the low concentrations of PolyB in brain tissue after administration in rats [23,34].

In the present study, signs of renal tubular damage, indicated by mild increases in the urinary GGT/creatinine ratio (25–100 IU/gCr), were found in 50% of horses and a more severe increase was found in one horse. Renal function impairment displayed by elevated serum creatinine values (1 horse) or elevated serum SDMA (2 horses) on the other hand was uncommon. Definition of acute renal failure is based on urinary output and a decrease in glomerular filtration rate displayed by an increase in serum creatinine over time [35,36]. In this study, we only examined renal function at one timepoint in most horses. Furthermore, most horses received fluid therapy during analysis precluding interpretation of some measurements such as urine specific gravity. Changes in renal function were therefore not thoroughly assessed and renal dysfunction might got underdiagnosed. Nephrotoxicity and associated renal failure is the major concern of polymyxin administration in human patients with an overall nephrotoxicity rate of 39% [17]. Accumulation and subsequent damage to renal tubular cells were demonstrated in rodent models and elevated markers of tubular damage might therefore reflect early nephrotoxic effects of this drug [34]. In contrast, previous studies in horses receiving PolyB showed no increases in the urinary GGT/creatinine ratio; however, some authors describe values up to 100 IU/gCr as normal [9,10,21]. Increases in the urinary GGT/creatinine ratio over 25 IU/gCr are considered elevated, but values between 25 and 100 IU/gCr are of unknown clinical relevance [35,37]. In the present study, pre-existing renal dysfunction due to the primary disease could not be differentiated from dysfunction due to nephrotoxicity of PolyB but seem likely since these were severely sick horses. Previous renal function impairment alters pharmacokinetics of PolyB and dose adaption is recommended in human patients [38]. The possible influence of changed drug kinetics by such effects in horses is unknown. Additionally, concomitant nephrotoxins such as aminoglycosides and non-steroidal anti-inflammatory drugs might have influenced results. For these reasons, a clear statement on the nephrotoxic effect of PolyB in hospitalized horses cannot be made based on our results. The nephrotoxic effect of polymyxins should, however, be considered in this vulnerable patient population.

Evaluation of neurological examinations is known to be highly rater dependent [39]. Similarly, inter-rater agreement was low in this study although three experienced observers were included. Analysis based on video recordings impeded the evaluation and lead to missing data as not all parts of the neurological exam were included in every video. A refined ataxia grading system, such as proposed in a previous study, would be desirable in a clinical setting to detect smaller differences over time or between horses [23]. However, due to missing data, it was not possible to apply this score in the present study.

Limitations of this study include the small sample size and therefore low power of statistical analyses. Originally, more horses were planned to be included but this study was prematurely finished as the used PolyB product was withdrawn from the market. The clinical nature of this study furthermore led to difficulties in standardization of examination and treatment of animals. While on one hand choosing a clinical population always leaves us with more heterogenous groups and more possible influence factors, on the other hand, these results from clinical patients will give us a more realistic information what to expect for our daily clinical work. The duration of treatment with polymyxin and other drugs was not standardized as these horses were clinical patients and treatment was therefore depending on their primary disease and the clinical response. Likewise, the use of additional medication was depending on several factors such as primary disease, disease severity, clinical signs, and laboratory results and might have also varied slightly depending on the clinician in charge. Missing neurological exams before treatment was started and incomplete neurological exams on video recordings were common. Analysis of pre-existing neurological signs and the use of a standardized ataxia score were therefore precluded. Furthermore, the descriptive nature of this study precludes any causative conclusion between the presence of PolyB and the development of clinical signs. Beside all limitations of this study, we still consider the results important for the equine clinicians and a good basis for larger more standardized clinical trials to fully understand the incidence, characteristics, severity, and duration of neurogenic and nephrogenic side effects of polymyxins in horses.

It is important to keep in mind that polymyxins are classified as critically important antimicrobial drugs in human medicine by the world health organization (WHO) [40]. Veterinarians using PolyB should therefore always weigh the use of this drug for the individual horse against one health considerations in addition to possible adverse effect.

## 4. Materials and Methods

This study was performed under the regulations of the Swiss federal authorities for animal experimentation (animal use license no. ZH 022/18).

### 4.1. Animals

Horses hospitalized at the Equine Hospital of the University of Zurich between March 2019 and February 2020 receiving PolyB and broad-spectrum antimicrobial therapy as part of their treatment plan were included in this study. Horses with diagnosed endotoxemia for which the clinician in charge prescribed PolyB as an antiendotoxemic treatment were prospectively enrolled. Clinical signs of endotoxemia in horses include elevated body temperature, tachypnea, tachycardia, and reddening of mucous membranes and laboratory findings include leucopenia or leucocytosis, toxic changes in neutrophils, and elevated inflammatory enzymes. Exclusion criteria were age younger than one year and PolyB or aminoglycoside treatment within the last 7 days before hospitalization and the absence of any video exams. Written owner consent was obtained. A sample size calculation determined that at least 18 horses per group are needed to detect a difference of 0.5 ataxia grade (standard deviation of 0.5 ataxia grade), with a power of 85% and an alpha error of 0.05.

### 4.2. Study Protocol

Horses were randomly allocated using a random number generator into one of two groups (https://www.randomizer.org/#randomize, accessed on 1 February 2019; *n* = 20 each). Depending on group allocation antimicrobial therapy was started. Group 1 (GENTA) received gentamicin (10 mg/kg bwt q24h IV, Genta 100 mg/mL, CP Pharma Schweiz AG Münchenstein) and penicillin (30,000 IU/kg bwt q6 h IV, Penicillin Natrium Streuli ad us. vet., Streuli Tiergesundheit AG Uznach), while group 2 (NO GENTA) received marbofloxacin (2 mg/kg bwt q24hr IV, Forcyl 160 mg/mL ad us. vet., Vetoquinol AG Bern) and penicillin (30,000 IU/kg q6hr IV). Horses in both groups received PolyB (6000 IU/kg q12hr IV, Polymyxin-B-sulfate 100,000 IU/mL compounded by the Kantonsapotheke Zurich, ingredients for 1 mL: 100,000 IU Polymyxin-B-sulfate, NaCl, Conserv. 1 mg E218). Antimicrobial and PolyB therapy were administered for as long as the clinician in charge of the therapy deemed necessary depending on clinical examination and laboratory findings. Concurrent medication was administered as deemed necessary by the clinician in charge of the case and recorded.

A full clinical and neurological examination was performed daily while the horses received PolyB and continued for an additional three days after discontinuation of PolyB treatment. The length of post-treatment monitoring was based on clinical experience and results in healthy horses where neurological signs resolved within three days after discontinuation of PolyB [23]. The neurological examination was performed 1–6 h after the first PolyB dose of the day and was recorded on video for later evaluation. Each video was numbered and randomized before analysis to mask treatment group and day to assure blinding of examiners. Evaluation of video sequences was performed individually by three examiners—one ECVN neurology diplomate, one ECEIM equine internal medicine diplomate, and one ACVIM large animal internal medicine diplomate.

Ataxia was graded using the ataxia scale 0–5 and presence of ataxia was defined as a median ataxia score ≥ 1/5 [41]. Observers were asked to grade deficits (none/mild/clear) in each point of the analysis (gait analysis at walk, with elevated head, backing, over obstacles, circling, tail pull, proprioception testing, the standing sway test, spinal reflex testing, and cutaneous sensation testing). Further, presence of weakness in general, during tail-pull and during the truncal sway testing was recorded. Weakness was defined as ≥2 observers agreeing upon the presence of weakness in at least one test. The used score sheet can be found in the Appendix A. The use of a previously described refined score including 21 ataxia grades was originally planned [23]. Due to missing parts of neurological exams on video recordings, this was only feasible in 4/75 exams and the score therefore not further evaluated in this group of horses.

Ataxia scores (0–5/5) are presented as medians based on ataxia scores of all three observers on video recordings. In addition, the median and range of ataxia scores of all horses and horses in both groups are shown. Maximum ataxia score was the highest median ataxia score in every horse during the study period.

PolyB serum concentrations [PolyB] were measured before the first dose of PolyB and daily 30 min after the first dose of the day. Once PolyB administration was discontinued, [PolyB] was measured every 24 h for an additional three days. Gentamicin plasma concentrations [Genta] were measured 30 min after each administration but not during the post-treatment period. Blood was taken from the catheter or by venipuncture once the catheter was removed. Blood was collected in serum and heparinized blood tubes, and serum and plasma harvested. All samples were stored at −80 °C until all samples were collected (sampling time March 2019 to February 2020) and measurements were performed all together (October 2021).

PolyB measurements were performed at the institute of forensic medicine of the University of Zurich using LC–MS (liquid chromatography–mass spectrometry). Serum samples (200 ul) were prepared by a simple protein precipitation procedure through the addition of a 3-fold volume of acetonitrile, shaking (5 min), centrifugation (10 min, 10,000 rpm), evaporation of 600 ul supernatant, and reconstitution in 120 uL mobile phase. Following analysis was performed by liquid chromatography–tandem mass spectrometry (LC–MS/MS) using a Thermo Fischer Ultimate 3000 UHPLC system (Thermo Fischer, San Jose, CA, USA) coupled to a Sciex 5500 QTrap linear ion trap quadrupole mass spectrometer (Sciex, Darmstadt, Germany). The LC settings were as follows: Phenomenex (Aschaffenburg, Germany) Synergi Polar-RP column (100 × 2.0 mm, 2.5 µm), gradient elution with 10 mM ammonium formate buffer in water containing 0.1% (*v*/*v*) formic acid (A) and acetonitrile containing 0.1% (*v*/*v*) formic acid (B). The flow rate was 0.5 mL/min with the following gradient: 0–0.5 min 2% B, 0.5–5.5 min to 60% B, 5.5–6 min to 98% B, hold at 98% B for 1.5 min and at 8.5 min re-equilibrating to 2% for 1.5 min. Retention times for Polymyxin B1, B2, and colistin B were 3.83 min, 3.61 min, and 3.42 min, respectively. For MS analysis, the following settings were applied: electrospray ionization (ESI) mode (Turbo V ion source; gas 1, nitrogen (50 psi); gas 2, nitrogen (60 psi); ion spray voltage, 4500; ion-source temperature, 550 °C; curtain gas, nitrogen (30 psi), collision gas, medium) and multiple reaction monitoring (MRM, quantifier and qualifier transitions for Polymyxin B1 602.6/120.1 (CE 45 eV), 602.6/100.9 (CE 30 eV), and 602.6/241.2 (CE 35 eV); Polymyxin B2 595.6/361.0 (CE 30 eV), 595.6/100.9 (CE 35 eV), and 595.6/233.2 (CE 40 eV); internal standard (IS) colistin B 578.5/227.3 (CE 30 eV), 578.5/100.9 (CE 30 eV), respectively; declustering potential (DP) 100, entrance potential (EP) 10, and collision exit potential (CXP) 12 V, respectively). Quantitative values were calculated against a six-point calibration curve (polymyxin concentrations of 0.05 ug/mL, 0.1 ug/mL, 0.5 ug/mL, 1 ug/mL, 2.5 ug/mL, 5 ug/mL; quadratic regression model, weighted 1/X) prepared in blank horse serum after summation of the peak areas of Polymyxin B1 and B2 over the peak area of colistin B as IS. The method was validated with four different QC concentrations (QC1 0.075 ug/mL, QC2 0.12 ug/mL, QC3 0.75 ug/mL, QC4 4.0 ug/mL, respectively) following a simplified method validation approach according to reference including selectivity, calibration model, the limit of quantification, matrix effects, accuracy/precision, and fulfilled all required criteria (selectivity: no interfering peaks; calibration: duplicate calibration with back-calculated concentrations within 20% of target; limit of quantification 0.05 ug/mL, with a signal-to-noise of 1:10; matrix effects in six serum samples at QC2 137% ± 16.6% and 107% ± 10% for Polymyxin B1 and B2, respectively; accuracy expressed as bias in % deviation of theoretical concentration and precision calculated as the relative standard deviation (RSD) in six replicates at QC1 15.4% (RSD 18.3%), QC2 2.5% (RSD 9.1%), QC3 2.7 (RSD 12.9%), QC4 4.3 (RSD 6.9%), respectively) [42]. All values below the limit of quantification were considered 0.00 μg/mL.

Gentamicin concentrations on heparinized plasma were measured by using fluorescence polarization on a COBAS INTEGRA 800 (Roche Diagnostics, Basel, Switzerland) by the Institute for Clinical Chemistry of the University of Zurich as previously described [43].

To assess kidney function, creatinine and urea were measured from heparinized plasma during PolyB treatment on day 1 and day 3, if available. On the same days (between 10 a.m. and 10 p.m.), urine analysis including dip stick analysis, sediment analysis, specific gravity, creatinine concentration, and GGT activity from a free catch urine sample were performed by the Clinical Laboratory of the University of Zurich. Creatinine, urea concentration, and GGT activity were measured photometrically on a Cobass 6000 (Roche Diagnostics, Basel, Switzerland) using the kinetic colorimetric compensated Jaffé method for creatinine concentration (Roche Creatinine Jaffé Gen.2), a coupled enzyme reaction (Roche UREAL) for urea concentration, and an enzymatic colorimetric method (Roche GGT2) for GGT activity. Urine dip stick analysis was performed with Combur10 dipsticks read with a Cobas u411 (Roche Diagnostics, Basel, Switzerland), specific gravity was determined by refractometry, and sediment analysis was performed with 10 mL of fresh urine. Symmetric dimethyl arginine (SDMA) was measured from frozen serum samples (−80 °C) taken on day 1 and measured in an external laboratory (Laboklin GmbH & Co, Bad Kissingen, Germany) by enzyme linked immunosorbent assay in October 2021.

Laboratory measurements including complete blood count with differential, clinical chemistry (bilirubin, urea, creatinine, symmetric dimethyl arginine (SDMA), protein, albumin, alkaline phosphatase (AP), aspartate aminotransferase (AST), γ-glutamyltransferase (GGT), glutamate dehydrogenase (GLDH), sorbitdehydrogenase (SDH), lactate dehydrogenase (LDH), creatinine kinase (CK), glucose, serum amyloid A (SAA), and fibrinogen), and electrolytes (sodium, kalium, chloride, calcium, magnesium, phosphate) were recorded when measured within the first 24 h after arrival. Systemic inflammatory response syndrome (SIRS) scoring was performed within the first 24 h of hospitalization based on changes in heart rate (>48 bpm), respiratory rate (>20 bpm), rectal temperature (>38.3 °C), and leukocytes (<4800/μL or >10,000/μL). A point was given for each criterium, with a maximum score of 4. SIRS was defined as present in horses with a score ≥2/4.

### 4.3. Statistical Analysis

Normality of the data was tested by the Shapiro–Wilk test. Due to non-normal distribution, data are presented as the median and range and non-normality was considered in the statistical analysis.

An ordinal mixed model fitted with the Laplace approximation was used to evaluate the effect of several factors on the daily and maximum ataxia scores of horses. Backward selection was used to retain significant explanatory variable. Horse number was included as random effect when evaluating the daily ataxia grade because repeated measures were used. Statistical analysis was undertaken in R (R Core Team 2020). Tolerated alpha error was set at 5%, with a *p* value < 0.05 considered statistically significant.

Following factors were analyzed on their effect on the daily ataxia grade: antibiotic treatment (GENTA vs. NO GENTA), number of PolyB doses, time since last PolyB dose (0.5 h during treatment, 24 h, 48 h, and 72 h on day post 1, post 2, and post 3, respectively), PolyB serum concentration, and gentamicin serum concentration if applicable. Factors analyzed on their effect on the maximum ataxia score were the duration of PolyB treatment, total doses of PolyB, administration of lidocaine (yes/no), plasma total protein concentration, plasma total protein concentration < 57 g/L (yes/no), plasma albumin concentration, plasma albumin concentration < 25 g/L (yes/no), serum amyloid A concentration, SIRS score (0–4), and presence of SIRS (yes/no).

Weighted kappa statistics were used to assess intra-rater agreement of ataxia gradings on six duplicates of videos using an online calculator (https://www.graphpad.com/quickcalcs/kappa1/, accessed on 31 November 2022). Inter-rater agreement using Krippendorff alpha analysis was measured between observers by including all analyzed videos (*n* = 81). Analysis was performed in R (R Core Team 2020).

The dataset will be made publicly available during review and provided prior to publication.

## 5. Conclusions

In conclusion, this study describes the occurrence of ataxia and weakness in hospitalized horses treated with PolyB as part of their treatment plan. Severity of ataxia decreased after discontinuation of PolyB treatment and a statistically significant effect of increasing time since the last dose on the ataxia grade was shown. Signs of tubular damage indicated by an elevated urinary GGT/creatinine ratio were frequent and nephrotoxic effects of PolyB should be considered when this drug is used in diseased horses.

## Figures and Tables

**Figure 1 antibiotics-12-00854-f001:**
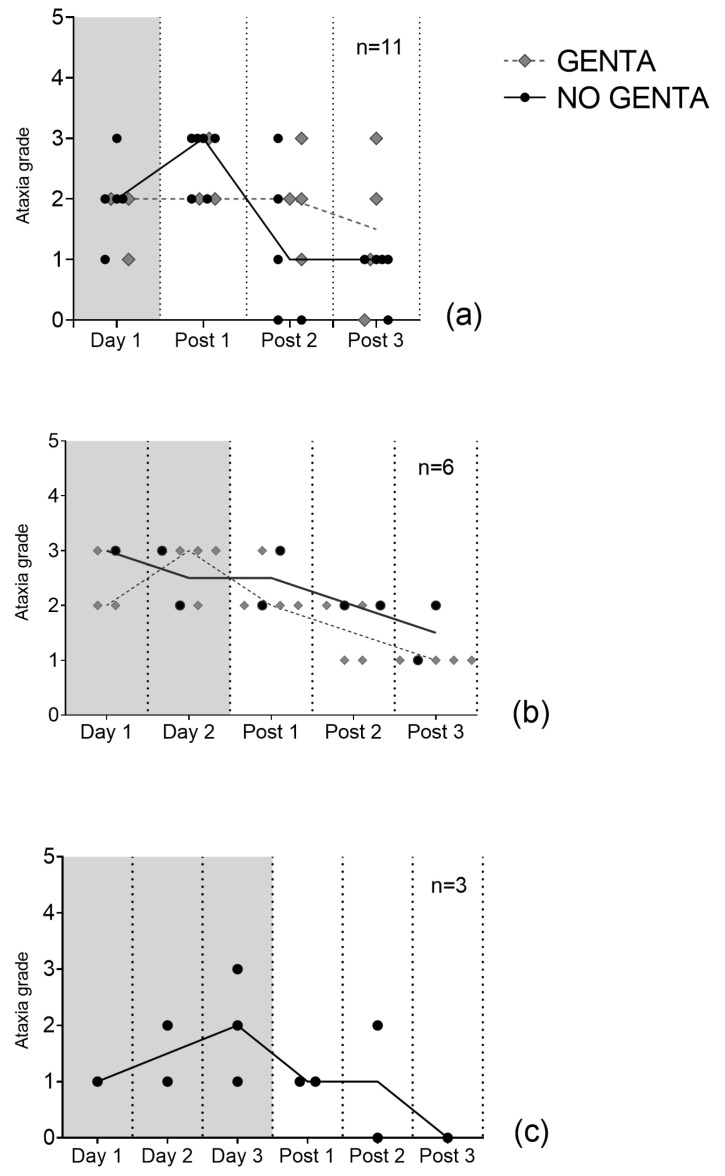
Median daily ataxia scores of 20 hospitalized horses receiving PolyB treatment (6000 IU/kg IV q12h). Horses received 1 day (**a**), 2 days (**b**), or 3 days (**c**) of PolyB treatment. Horses were additionally treated with GENTA (gentamicin/penicillin) or NO GENTA (marbofloxacin/penicillin) as antimicrobial treatment. Presented are the median scores of three blinded observers’ gradings on video exams. Lines represent the median of all horses’ ataxia grades.

**Figure 2 antibiotics-12-00854-f002:**
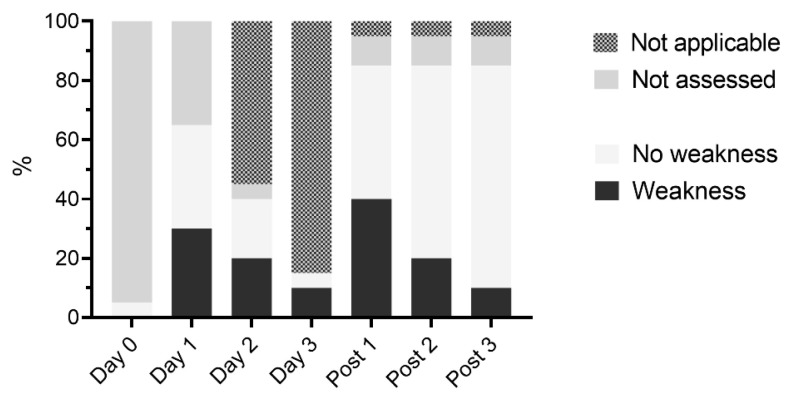
Presence of weakness in hospitalized horses receiving PolyB treatment (6000 IU/kg IV q12h). Horses received 1 day (*n* = 11), 2 days (*n* = 6), or 3 days (*n* = 3) of PolyB treatment. Neurological exam was performed daily and three additional days (post 1–3) and analyzed on video recordings by three blinded observers.

**Figure 3 antibiotics-12-00854-f003:**
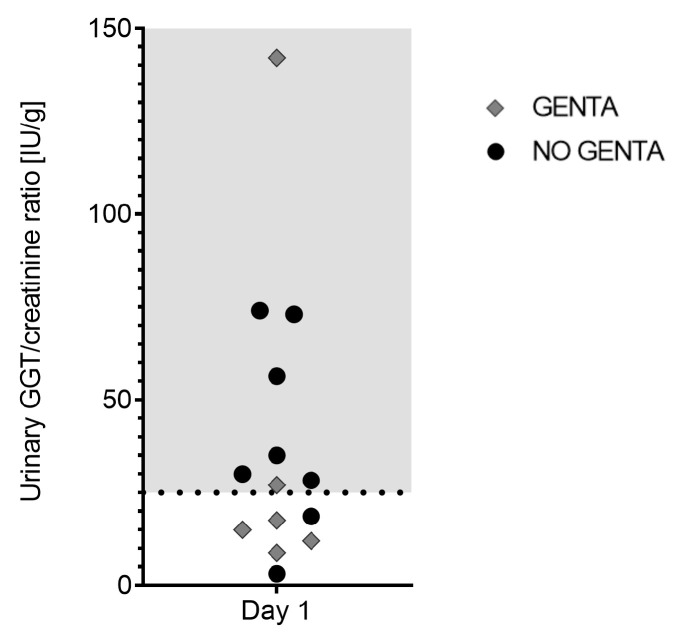
Urinary GGT/creatinine ratio [IU/g] in 14 hospitalized horses on day 1 of receiving PolyB treatment (6000 IU/kg IV q12h). Horses were additionally treated with GENTA (gentamicin/penicillin) or NO GENTA (marbofloxacin/penicillin) as antimicrobial treatment. Shaded area indicates elevated values (>25 IU/g).

**Figure 4 antibiotics-12-00854-f004:**
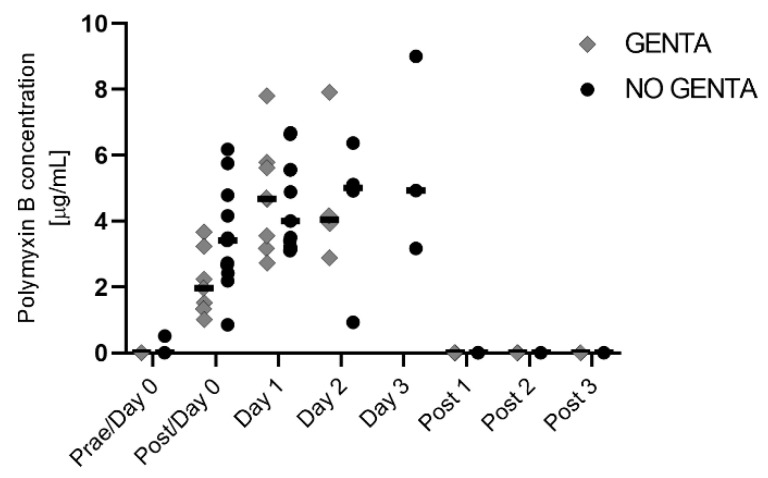
PolyB serum concentration in hospitalized horses receiving PolyB treatment (6000 IU/kg IV q12h). Horses received 1 day (*n* = 11), 2 days (*n* = 6), or 3 days (*n* = 3) of PolyB treatment. Horses were additionally treated with GENTA (gentamicin/penicillin) or NO GENTA (marbofloxacin/penicillin) as antimicrobial treatment. Serum was taken 30 min after the first dose of the day and was performed daily during treatment and three additional days (post 1–3). Individual values of horses are shown, horizontal lines indicate the median.

## Data Availability

The dataset will be made publicly available during review and provided prior to publication.

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
