# Peer review of "Preliminary Investigation of Side Effects of Polymyxin B Administration in Hospitalized Horses"

_antibiotics, 2023, doi:10.3390/antibiotics12050854_

Round 1
Reviewer 1 Report
The submitted article investigates the possible side effects of polymixin B administration in horses and covers a scientific gap of the literature.
The article itself is well written but the experimental design has some flaws which are also stated by the authors. The main problem consists in the minimal sample size of 18 horses per group that has not been achieved; for this particular reason statistical inference cannot be used to transfer the obtained results in the observed population to the entire population. Given this considerations I would suggest to the authors to change the title of the present manuscript with something along the lines of "preliminary investigation of...".
Nevertheless the present work has great scientific soundness especially for hippiatry and the authors should consider to establish a new experimental design with appropriate statistical power in order to continue this investigation, especially given the results obtained in this work.
Author Response
Reviewer 1
The submitted article investigates the possible side effects of polymixin B administration in horses and covers a scientific gap of the literature.
The article itself is well written but the experimental design has some flaws which are also stated by the authors. The main problem consists in the minimal sample size of 18 horses per group that has not been achieved; for this particular reason statistical inference cannot be used to transfer the obtained results in the observed population to the entire population. Given this considerations I would suggest to the authors to change the title of the present manuscript with something along the lines of "preliminary investigation of...".
Answer of the author: Thank you for your time and effort in revising this manuscript. We do agree with this comment and changed the title of the manuscript.
Nevertheless the present work has great scientific soundness especially for hippiatry and the authors should consider to establish a new experimental design with appropriate statistical power in order to continue this investigation, especially given the results obtained in this work.
Reviewer 2 Report
In this clinical study, the authors aimed to describe the incidence, characteristics, severity, and duration of neurogenic and nephrogenic side effects after PolyB administration in hospitalized horses.
The subject of the study is original and falls within the scope of the journal. The authors present a well-written manuscript describing a well-organized study. Moreover, I think this investigation provides valuable data for scientific literature and the readers of the journal.
1. P3: “ROTA TS”? The abbreviations should be explained the first time it appears in the text and then can be used throughout the manuscript.
2. The number of animals included in the study is small. Moreover, there may be an interaction between polymyxin and other concomitant drugs and this has been ignored.
3. P12: The details of the analytical method for the determination of Poly B in plasma are not enough. Which method was used for the analysis? Citation? Has this method been validated? Information about plasma extraction is not provided in the manuscript. In addition, the detection limit and quantification limits should be given. The limit of quantification, not detection, is acceptable in chromatographic analysis.
4. Reference list: Please check the consistency of the reference list for the guidelines of the journal (most of the journal names should be revised).
Author Response
Reviewer 2
In this clinical study, the authors aimed to describe the incidence, characteristics, severity, and duration of neurogenic and nephrogenic side effects after PolyB administration in hospitalized horses.
The subject of the study is original and falls within the scope of the journal. The authors present a well-written manuscript describing a well-organized study. Moreover, I think this investigation provides valuable data for scientific literature and the readers of the journal.
Answer of the author: Thank you for your time and effort in revising this manuscript.
P3: “ROTA TS”? The abbreviations should be explained the first time it appears in the text and then can be used throughout the manuscript.
Answer of the authors: ROTA TS is a product name not an abbreviation. It is the product name and the ingredient is trimethoprim sulfonamides.
The number of animals included in the study is small. Moreover, there may be an interaction between polymyxin and other concomitant drugs and this has been ignored.
Answer of the authors: We do not agree with this comment. We are aware of possible interactions with several drugs concomitantly used. We specifically look for an interaction with gentamicin by creating two different groups. We are also aware of other drugs influencing our results. Lidocaine was taken as factor into statistical analysis, but did not reveal significant, however, the power of this analysis is low. The possible influence of other drugs with known effect on the neurological system are discussed in a paragraph (L296-335).
3. P12: The details of the analytical method for the determination of Poly B in plasma are not enough. Which method was used for the analysis? Citation? Has this method been validated? Information about plasma extraction is not provided in the manuscript. In addition, the detection limit and quantification limits should be given. The limit of quantification, not detection, is acceptable in chromatographic analysis.
Answer of the authors: Details on the method used were inserted in the manuscript in the methodology section (L504-550).
4. Reference list: Please check the consistency of the reference list for the guidelines of the journal (most of the journal names should be revised).
Answer of the authors: The Endnote output style provided by the publisher (https://endnote.com/style_download/mdpi/) was used for all references.
Reviewer 3 Report
This manuscript refers to an investigation on the side effects of the administration of polymyxin B in hospitalized horses. The study is interesting, from the clinical point of view, since it seeks to establish an assessment of the use of polymyxin B in horses; however, the manuscript is disordered, in addition to the fact that the authors describe a series of antibiotics used during the investigation that can generate undesirable interactions or indiscriminate use of them. In addition, the authors suggest that their results (development of ataxia and weakness) may be associated with the co-administration of gentamicin and the number of doses administered as risk factors, which cannot be fully demonstrated due to the age of the horses, since young animals have a greater volume of distribution than adults and also, to the drug interactions that occur with all the drugs that were used in the study, for example, sulfas with trimethoprim, metronidazole, flunixin, meloxicam, lidocaine, among others. A thorough review of the experimental design is also required, even though statistical analysis could be a strength of the research. The main weakness identified is that the descriptive nature of this study precludes any causal conclusion between the presence of PolyB and the development of clinical signs, due to multifactorial causes and variables that have an influence on its design.
Regarding the review format, authors are asked to follow the instructions guide, so that the line numbers appear on each page, but above all the sections in the order in which they are requested in said guide. This makes it easier for reviewers to review and provide feedback.
General comments on the manuscript
Please consider adding the study aim to your abstract. Likewise, modify the conclusion in this section and verify that it is consistent with the aim of the investigation.
Introduction
Add studies that expose the nephrotoxicity and neurotoxicity of polymyxin B in horses, to avoid comparing with the findings described in humans.
The materials and methods section does not justify the variable duration of administration of polymyxin B and concurrent medication with other drugs. Likewise, the duration of the videotaping of the neurological examinations is not specified, nor is the moment in which they were performed.
The ataxia scale and weakness score used to require a more detailed description and references to support their use.
Some used equipment lacks brand, model, or country of origin.
The methods described in this section of the manuscript must be supported by at least one reference.
To assess renal function, heparinized plasma creatinine, and urea were measured during PolyB treatment on day 1 and day 3. If available? Please, clarify
In statistical analysis, delete the phrase: “and visual inspection of the data”.
The results include data that could be placed in the methodology section.
The inclusion and exclusion criteria are not correctly specified. For example, this question: do the different health states of the animals not affect the neurological evaluations or the presentation of ataxia or weakness?
How can they influence the results and what drug interactions can be observed with these drugs that were used in some animals? Flunixin (18/20, 90 %), dalteparin (15/20, 75 %), lidocaine (CRI up to 24 h duration, 5/20, 25 % ), omeprazole (3/20, 15 %), heparin (2/20, 10 %), metamizole (2/20, 10 %), sucralfate (2/20, 10 %), fenbendazole (1/20, 5 % ), pergolide (1/20, 5 %), meloxicam (1/20, 5 %), sedative drugs including xylazine (9/20, 45 %), detomidine (4/20, 20 %) y butorphanol (6/20, 30 %).
The authors mention that neurological examinations were performed at least 2 hours after the administration of sedative drugs. It's right? If so, there may be a bias or influence of drug residues in the body, which would directly affect the results. In his case, it is also mentioned that these evaluations were carried out daily and up to three days after the last dose with polymyxin B, which in the case of gentamicin would make it possible to study and establish a risk factor, but not with the rest of the drugs.
Up to table 2, the dose of polymyxin B used is mentioned, however, the reason or foundation for the different duration of the treatments with this antibiotic continues to be unknown.
The presentation of results in table 1 and figure 1 is repetitive in information and content.
I suggest the authors calculate other risk factors, not just co-administration with gentamicin. For example, the sex and age of the animal, body condition or weight, concomitant disease, and even the co-administration of other drugs.
The authors mention that the concordance between evaluators is fair to good, which is uncertain since according to Krippendorff, a result of 0.667 would be the acceptable lower limit and a result ≥ 0.800 would be acceptable. In the present study, a value of 0.59 is presented, so an admissible value is not reached.
Modify the way of presenting the descriptive statistics in Figure 2 as it causes confusion and error when interpreting the results. For example, on day 3 66% of horses with weakness are observed but this percentage corresponds to 2 individuals, compared to day 1 where 46% is observed with weakness, however, this is represented by 6 animals, which is a higher number.
Renal function tests were measured at different times for all the horses and under various conditions, which is why it does not allow for correctly establishing the nephrotoxic effect of polymyxin B with or without co-administration of gentamicin. For example, urine-specific gravity was measured when animals received fluid therapy and were not a repeated parameter at any other time.
Regarding the serum concentration of polymyxin, the Cmax and half-life of this antibiotic in a healthy animal are 30 min and 6 h on average, respectively, while the half-life with renal failure ranges from 48-72 h. This pharmacokinetic behavior in the distribution of the antibiotic is a factor that is not discussed or considered in the design of the study or the corresponding sections.
The discussion and conclusions must be rewritten according to the modification of the manuscript.
Finally, please add the registration number and approval by an ethics committee for the use of animals.
Author Response
Reviewer 3
This manuscript refers to an investigation on the side effects of the administration of polymyxin B in hospitalized horses. The study is interesting, from the clinical point of view, since it seeks to establish an assessment of the use of polymyxin B in horses; however, the manuscript is disordered, in addition to the fact that the authors describe a series of antibiotics used during the investigation that can generate undesirable interactions or indiscriminate use of them. In addition, the authors suggest that their results (development of ataxia and weakness) may be associated with the co-administration of gentamicin and the number of doses administered as risk factors, which cannot be fully demonstrated due to the age of the horses, since young animals have a greater volume of distribution than adults and also, to the drug interactions that occur with all the drugs that were used in the study, for example, sulfas with trimethoprim, metronidazole, flunixin, meloxicam, lidocaine, among others. A thorough review of the experimental design is also required, even though statistical analysis could be a strength of the research. The main weakness identified is that the descriptive nature of this study precludes any causal conclusion between the presence of PolyB and the development of clinical signs, due to multifactorial causes and variables that have an influence on its design.
Answer of the authors: Thank you for your time and effort in revising this manuscript.
Regarding the review format, authors are asked to follow the instructions guide, so that the line numbers appear on each page, but above all the sections in the order in which they are requested in said guide. This makes it easier for reviewers to review and provide feedback.
Answer of the authors: This was checked and adapted as suggested. The needed order of sections is not clear from the instruction guidelines (Methodology after Introduction or after Discussion), but the order was made according to most recent original articles in this journal.
General comments on the manuscript
Please consider adding the study aim to your abstract. Likewise, modify the conclusion in this section and verify that it is consistent with the aim of the investigation.
Answer of the authors: The aim of the study was added to the abstract.
Introduction
Add studies that expose the nephrotoxicity and neurotoxicity of polymyxin B in horses, to avoid comparing with the findings described in humans.
Answer of the authors: There is very limited data in horses, therefore we chose to compare it to human literature in addition. Available studies in horses are included in an own paragraph (L67-78).
The materials and methods section does not justify the variable duration of administration of polymyxin B and concurrent medication with other drugs. Likewise, the duration of the videotaping of the neurological examinations is not specified, nor is the moment in which they were performed.
Answer of the authors: The aim of this study was to describe the adverse effects in clinical horse patients with endotoxemia. While on one hand choosing a clinical population always leaves us with more heterogenous groups and more possible influence factors, on the other hand these results from clinical patients will give us a more realistic information what to expect for our daily clinical work. The duration of treatment with polymyxin and other drugs was not standardized as these horses were clinical patients and treatment was therefore depending on their primary disease and the clinical response. Likewise, the use of additional medication was depending on several factors such as primary disease, disease severity, clinical signs, and laboratory results and might have also varied slightly depending on individual clinician in charge. In general, polymyxin B was stopped depending on improvement of clinical signs and of improvement of hematology and biochemistry signs of endotoxemia. This information was added in L453. Time points of neurological examinations were added in L462. Duration of videotaping was not specifically noted but included the full time of the neurological examination which generally takes approximately 10 minutes.
The ataxia scale and weakness score used to require a more detailed description and references to support their use.
Answer of the authors: A reference was given for the ataxia scoring, which is a universally used and established grading system in horses. Lack of inter-rater agreement of ataxia scoring in general is discussed (L396). Weakness was not scored but only considered present or absent when 2/3 observers agreed to its presence/absence.
Some used equipment lacks brand, model, or country of origin.
Answer of the authors: Missing information was added.
The methods described in this section of the manuscript must be supported by at least one reference.
Answer of the authors: We assume this refers to the measurements of polymyxin and gentamicin. Additional information and references were included here (L504-554).
To assess renal function, heparinized plasma creatinine, and urea were measured during PolyB treatment on day 1 and day 3. If available? Please, clarify
Answer of the authors: This refers to some missing data (see L201). This information was deleted in the methods section to not confuse the reader.
In statistical analysis, delete the phrase: “and visual inspection of the data”.
Answer of the authors: This was changed as suggested.
The results include data that could be placed in the methodology section.
Answer of the authors: We are unaware of data in the results section which should be placed into the methodology section. Could you please point out specifically witch data you are referring if needed.
The inclusion and exclusion criteria are not correctly specified. For example, this question: do the different health states of the animals not affect the neurological evaluations or the presentation of ataxia or weakness?
Answer of the authors: Inclusion criteria were solely the administration of PolyB and antibiotics. Exclusion criteria were age younger than one year and PolyB or aminoglycoside treatment within the last 7 days before hospitalization. Originally pre-treatment examination (day 0) was planned to assess preexisting neurological abnormalities, however, included patients usually presented after hours and in an emergency setting for non-neurological disease, such that neurological exam could almost never be performed (n=1/20). Pre-existing neurologic disease can therefore not be excluded, however is not common in horses as this is a big safety factor and affected horses cannot be handled or ridden safely so that owners will report upon neurological conditions immediately. Disease status could also affect the neurological examination. We therefore included factors such as systemic inflammatory reaction scoring and protein concentrations to find possible relationships between disease status and the severity of neurological abnormalities. These influencing factors are discussed in L280 and L337.
How can they influence the results and what drug interactions can be observed with these drugs that were used in some animals? Flunixin (18/20, 90 %), dalteparin (15/20, 75 %), lidocaine (CRI up to 24 h duration, 5/20, 25 % ), omeprazole (3/20, 15 %), heparin (2/20, 10 %), metamizole (2/20, 10 %), sucralfate (2/20, 10 %), fenbendazole (1/20, 5 % ), pergolide (1/20, 5 %), meloxicam (1/20, 5 %), sedative drugs including xylazine (9/20, 45 %), detomidine (4/20, 20 %) y butorphanol (6/20, 30 %).
Answer of the authors: The possible effect of additional drugs is discussed already but the section was extended, and a new reference inserted (L296-335).
The authors mention that neurological examinations were performed at least 2 hours after the administration of sedative drugs. It's right? If so, there may be a bias or influence of drug residues in the body, which would directly affect the results. In his case, it is also mentioned that these evaluations were carried out daily and up to three days after the last dose with polymyxin B, which in the case of gentamicin would make it possible to study and establish a risk factor, but not with the rest of the drugs.
Answer of the authors: We do agree with this statement. The drugs used for sedation (xylazine, detomidine) have a short elimination half time of <1h and clinical effects are not seen after this time. Concurrent treatment with these or other medications had an unknown influence on the results, which is a major limitation of our study. However, this was inevitable due to the clinical nature of the trial including sick horses in need of other medication. We intensively discuss the influence of concurrent drugs (L296-335).
Up to table 2, the dose of polymyxin B used is mentioned, however, the reason or foundation for the different duration of the treatments with this antibiotic continues to be unknown.
Answer of the authors: Polmyxin B was used as an antiendotoxemic treatment in these horses and was given as long as signs of endotoxemia were present. Signs of endotoxemia in horses include clinical signs (elevated body temperature, tachypnea, tachycardia, reddening of mucous membranes, etc.) and laboratory findings (leucopenia or leucocytosis, toxic changes of neutrophils, elevated inflammatory enzymes, etc.). This information was added in L453.
The presentation of results in table 1 and figure 1 is repetitive in information and content.
Answer of the authors: Table 1 was deleted from the manuscript and is provided as supplementary information.
I suggest the authors calculate other risk factors, not just co-administration with gentamicin. For example, the sex and age of the animal, body condition or weight, concomitant disease, and even the co-administration of other drugs.
Answer of the authors: We do agree that including further risk factors would be interesting. However, due to the low number of animals, evaluation of further factors would only have low power and the more factors are included the lower the power of the analysis gets. We therefore considered selected risk factors based on our clinical experience (all ages and sexes are affected by these side effects from clinical experience, PolyB is dosed for body weight, and gentamicin is a commonly used drug with known neuromuscular blocking effect) carefully and do not think adding more factors will improve the study.
The authors mention that the concordance between evaluators is fair to good, which is uncertain since according to Krippendorff, a result of 0.667 would be the acceptable lower limit and a result ≥ 0.800 would be acceptable. In the present study, a value of 0.59 is presented, so an admissible value is not reached.
Answer of the authors: We changed the sentence in L187. Low rater agreement of ataxia scorings is discussed in L396.
Modify the way of presenting the descriptive statistics in Figure 2 as it causes confusion and error when interpreting the results. For example, on day 3 66% of horses with weakness are observed but this percentage corresponds to 2 individuals, compared to day 1 where 46% is observed with weakness, however, this is represented by 6 animals, which is a higher number.
Answer of the authors: Figure 2 was adjusted, and data of all horses included at all timepoint.
Renal function tests were measured at different times for all the horses and under various conditions, which is why it does not allow for correctly establishing the nephrotoxic effect of polymyxin B with or without co-administration of gentamicin. For example, urine-specific gravity was measured when animals received fluid therapy and were not a repeated parameter at any other time.
Answer of the authors: We do agree to this. We adapted the discussion section and added an additional sentence on this (L352-395).
Regarding the serum concentration of polymyxin, the Cmax and half-life of this antibiotic in a healthy animal are 30 min and 6 h on average, respectively, while the half-life with renal failure ranges from 48-72 h. This pharmacokinetic behavior in the distribution of the antibiotic is a factor that is not discussed or considered in the design of the study or the corresponding sections.
Answer of the authors: Thank you for this comment. This consideration was added in the discussion and an additional reference added.
The discussion and conclusions must be rewritten according to the modification of the manuscript.
Answer of the authors: Changes are made as stated in previous points.
Finally, please add the registration number and approval by an ethics committee for the use of animals.
Answer of the authors: This was added on L424.
Round 2
Reviewer 3 Report
I am grateful that the author has taken into account my comments in the first revision of his manuscript. It seems to me that the article has improved substantially, however, there are still some aspects that require the attention of the authors, which I consider must be discussed, in order to achieve a quality publication.
L32: please indicate the meaning of the abbreviation GGT. Do the same on the L68.
L36-37: the conclusion should be clearer regarding the observed nephrogenic effects, it is not enough to explain that these effects should be monitored. In fact, the aim described in L80-82 includes the description of incidence, characteristics, severity, and duration, which do not appear in its conclusion. Please rewrite.
L91-93: The authors should be more clear about the inclusion and exclusion criteria. Please, elaborate more on this, however, this correction I suggest to the authors be considered in L411-414 in materials and methods, point 4.1. In your response to this reviewer, you mention some extra criteria that are not described in your manuscript. Even in the L334-336, there are some other criteria that could be indicated as exclusion criteria.
L123-131: drug interactions with omeprazole, heparin, metamizole, and sucralfate (drugs listed in these lines), require further discussion in L323-328.
The authors mention that neurological examinations were performed at least 2 hours after the administration of sedative drugs. In the response issued by the authors it is recognized that this was a limitation of the study, therefore it should be discussed as such, before the conclusions.
L486: please provide centrifugation constants
L554-555: please clarify the analytes or parameters obtained in the complete blood count and serum chemistry. Moreover, indicate that they are found in supplementary material 2.
Finally, please include this explanation provided in your response to this reviewer in the discussion section so that the reader can understand that this is a preliminary study:
“The aim of this study was to describe the adverse effects in clinical horse patients with endotoxemia. While on one hand choosing a clinical population always leaves us with more heterogenous groups and more possible influence factors, on the other hand, these results from clinical patients will give us a more realistic information what to expect for our daily clinical work. The duration of treatment with polymyxin and other drugs was not standardized as these horses were clinical patients and treatment was therefore depending on their primary disease and the clinical response. Likewise, the use of additional medication was depending on several factors such as primary disease, disease severity, clinical signs, and laboratory results and might have also varied slightly depending on the individual clinician in charge. In general, polymyxin B was stopped depending on improvement of clinical signs and of improvement of hematology and biochemistry signs of endotoxemia”.
Once it's included, dig a little deeper into explaining those factors.
Author Response
I am grateful that the author has taken into account my comments in the first revision of his manuscript. It seems to me that the article has improved substantially, however, there are still some aspects that require the attention of the authors, which I consider must be discussed, in order to achieve a quality publication.
Answer of the authors: Thank you for your work and helping in improving our manuscript. We have addressed all your further concerns and we hope that we have been able to answer all outstanding questions to your satisfaction. Please find our detailed answers below.
L32: please indicate the meaning of the abbreviation GGT. Do the same on the L68.
Answer of the authors: This has been changed as requested.
L36-37: the conclusion should be clearer regarding the observed nephrogenic effects, it is not enough to explain that these effects should be monitored. In fact, the aim described in L80-82 includes the description of incidence, characteristics, severity, and duration, which do not appear in its conclusion. Please rewrite.
Answer of the authors: We specified the conclusions concerning nephrotoxic effects as you suggested.
L91-93: The authors should be more clear about the inclusion and exclusion criteria. Please, elaborate more on this, however, this correction I suggest to the authors be considered in L411-414 in materials and methods, point 4.1. In your response to this reviewer, you mention some extra criteria that are not described in your manuscript. Even in the L334-336, there are some other criteria that could be indicated as exclusion criteria.
Answer of the authors: We included presence of endotoxemia as the indication of the use of PolyB in these horses. We made clear that this was a decision made by the responsible clinician and was not part of the study. We also included the absence of any video recordings as an exclusion criterion in the Material section as already listed in the Results section (L429). You mention L334-336 – we cannot exactly understand which criteria you mean there. The presence of preexisting neurological abnormalities was not an exclusion criteria per se, but we would have liked to have a baseline neurologic examination to detect any of such. However, as we stated in the last revision notes, this was rarely done due to the emergency settings and neurologic abnormalities are not common in horses as this is a big safety factor and affected horses cannot be handled or ridden safely so that owners will report upon neurological conditions immediately.
L123-131: drug interactions with omeprazole, heparin, metamizole, and sucralfate (drugs listed in these lines), require further discussion in L323-328.
Answer of the authors: We have further extended the discussion on drug interactions and included two additional references. As suggested we highlighted the unpredictable influence of these interactions and specifically mention it as a major limitation (L331-341).
The authors mention that neurological examinations were performed at least 2 hours after the administration of sedative drugs. In the response issued by the authors it is recognized that this was a limitation of the study, therefore it should be discussed as such, before the conclusions.
Answer of the authors: This was discussed in the section on possible drug interactions and specifically mentioned as a limitation (L321-341).
L486: please provide centrifugation constants
Answer of the authors: This was included as requested.
L554-555: please clarify the analytes or parameters obtained in the complete blood count and serum chemistry. Moreover, indicate that they are found in supplementary material 2.
Answer of the authors: Obtained parameters were included as requested in L593. Link to the supplementary material is already included in the Results section in L136.
Finally, please include this explanation provided in your response to this reviewer in the discussion section so that the reader can understand that this is a preliminary study:
“The aim of this study was to describe the adverse effects in clinical horse patients with endotoxemia. While on one hand choosing a clinical population always leaves us with more heterogenous groups and more possible influence factors, on the other hand, these results from clinical patients will give us a more realistic information what to expect for our daily clinical work. The duration of treatment with polymyxin and other drugs was not standardized as these horses were clinical patients and treatment was therefore depending on their primary disease and the clinical response. Likewise, the use of additional medication was depending on several factors such as primary disease, disease severity, clinical signs, and laboratory results and might have also varied slightly depending on the individual clinician in charge. In general, polymyxin B was stopped depending on improvement of clinical signs and of improvement of hematology and biochemistry signs of endotoxemia”.
Answer of the authors: We included parts of this explanation and added a sentence to support the preliminary nature of this study (L404-427).
Once it's included, dig a little deeper into explaining those factors.
Answer of the authors: We are not sure about the exact meaning of this comment. We extended the discussion on possible influencing factors (L324, L351, L404) and also included explanations.